# MicroRNAs as Biomarkers of Surgical Outcome in Mesial Temporal Lobe Epilepsy: A Systematic Review

**DOI:** 10.3390/ijms24065694

**Published:** 2023-03-16

**Authors:** Alexey M. Yakimov, Elena E. Timechko, Irina G. Areshkina, Anna A. Usoltseva, Kristina D. Yakovleva, Elena A. Kantimirova, Nikita Utyashev, Nikita Ivin, Diana V. Dmitrenko

**Affiliations:** 1Department of Medical Genetics and Clinical Neurophysiology of Postgraduate Education, V.F. Voino-Yasenetsky Krasnoyarsk State Medical University, 660022 Krasnoyarsk, Russia; 2Federal State Budgetary Institution “National Medical and Surgical Center Named after N.I. Pirogov”, 105203 Moscow, Russia

**Keywords:** temporal lobe epilepsy, microRNA, biomarkers, surgery, outcome

## Abstract

Mesial temporal lobe epilepsy is the most common type of epilepsy. For most patients suffering from TLE, the only treatment option is surgery. However, there is a high possibility of relapse. Invasive EEG as a method for predicting the outcome of surgical treatment is a very complex and invasive manipulation, so the search for outcome biomarkers is an urgent task. MicroRNAs as potential biomarkers of surgical outcome are the subject of this study. For this study, a systematic search for publications in databases such as PubMed, Springer, Web of Science, Scopus, ScienceDirect, and MDPI was carried out. The following keywords were used: temporal lobe epilepsy, microRNA, biomarkers, surgery, and outcome. Three microRNAs were studied as prognostic biomarkers of surgical outcome: miR-27a-3p, miR-328-3p, and miR-654-3p. According to the results of the study, only miR-654-3p showed a good ability to discriminate between patients with poor and good surgical outcomes. MiR-654-3p is involved in the following biological pathways: ATP-binding cassette drug transporters, glutamate transporter SLC7A11, and TP53. A specific target for miR-654-3p is GLRA2, the glycine receptor subunit. MicroRNAs, which are diagnostic biomarkers of TLE, and epileptogenesis, miR-134-5p, MiR-30a, miRs-143, etc., can be considered as potential biomarkers of surgical outcome, as they can be indicators of early and late relapses. These microRNAs are involved in the processes characteristic of epilepsy: oxidative stress and apoptosis. The study of miRNAs as potential predictive biomarkers of surgical outcome is an urgent task and should be continued. However, when studying miRNA expression profiles, it is important to take into account and note a number of factors, such as the type of sample under study, the time of sampling for the study, the type and duration of the disease, and the type of antiepileptic treatment. Without taking into account all these factors, it is impossible to assess the influence and involvement of miRNAs in epileptic processes.

## 1. Introduction

Mesial temporal lobe epilepsy (mTLE) is a heterogeneous disease characterized by the localization of the epileptogenic focus in the temporal mesial structures (hippocampus, entorhinal cortex, amygdala, and parahippocampal gyrus) [1,2]. Hippocampal sclerosis (HS) is the most common pathological substrate in TLE [3]. mTLE occurs in one-third of patients [3,4]. However, 25–30% of patients develop drug-resistance. Currently, in two-thirds of patients, the only way to achieve freedom from epileptic seizures is a surgical approach [5], but in this case there is a 51% chance of relapse [6,7].

Regardless of the type of surgery, relapses are divided into two possible outcomes: post-surgical seizures appear 2–6 months after surgery or over the next 10–15 years. As a result, it becomes necessary to divide relapses into “early” and “late”, respectively. The first type is characterized by inaccuracy in determining the epileptogenic focus, or its incomplete resection, while late relapses are characterized by the appearance of a new epileptic focus [6].

Such temporal patterns are characteristic of all types of surgical treatment of epilepsy and, in the concrete, do not depend on the type, location, and pathology of epilepsy [8].

The occurrence of “early” relapse is a continuation of epileptic activity at the site of the initial focus. Relapse in the first 6 months after surgery leads to the development of drug-resistant epilepsy four times more often [9].

In the case of late relapses, seizures occur less frequently, they are more likely to be treated with anticonvulsants, and the presence of a pathological substrate of the resected tissue is also not observed [9,10]. In their pathogenetic features, late relapses resemble first-time epilepsy. Late relapses may be predicted by comorbidities of epilepsy, such as focal cortical dysplasia type one [11,12,13].

Electroencephalography (EEG) is the standard method for diagnosing seizures. However, in some cases, routine EEG does not provide a satisfactory result due to the absence of lesions in the temporal lobe or in the presence of extratemporal epilepsy. Therefore, for pre-surgical analysis, the invasive EEG method is used, which makes it possible to place electrodes on the surface or in the substance of the brain [14]. The advantage of this method is the ability to accurately identify the seizure area, as well as predicting a good outcome of surgical treatment. However, this method has a number of significant drawbacks: the complexity of the procedure, since surgical intervention is required; limited coverage of the studied area of the brain; and the impossibility of predicting late relapses [15]. Therefore, the search for new diagnostic methods, the use of which would reduce the degree of invasiveness and increase the accuracy of the prognosis, is relevant [14].

Epilepsy biomarkers can be divided into two large groups: diagnostic ones, which provide information about the clinical status, and prognostic ones, which make it possible to predict the development of future clinical signs, as well as the possibility of remission or relapse [16]. In general, biomarkers include EEG (electroencephalogram of the brain), neuroimaging methods (computed tomography, magnetic resonance imaging), genetic tests, and metabolism analysis [17]. MicroRNAs are promising both as a diagnostic and as a prognostic molecular biomarker [18].

There are no descriptions of validated markers in the publications that could contribute to an unambiguous determination of the surgical prognosis before resection of the epileptogenic zone [19,20], but research in this area is necessary and the search for such markers continues [21,22].

MicroRNAs are promising as both diagnostic and prognostic molecular biomarkers of mTLE [18,23]. They are less invasive, since only blood sampling is required to perform the necessary research [24]. MicroRNAs are extremely stable under conditions that typically result in the degradation of most RNAs, such as high or low temperatures and changes in the acidity of the environment. Such stability is due to their bonds with proteins and lipoprotein complexes [25].

## 2. Materials and Methods

The analysis of publications in the databases PubMed, Springer, Web of Science, Scopus, Springer, ScienceDirect, and MDPI was carried out. The search was performed using the keywords: temporal lobe epilepsy, miRNA, biomarker, and surgical outcome. Publications from 2012 to 2023 were analyzed, including original research and review articles. Bioinformatic analysis was carried out using such platforms as: MiRTarBase, ShinyGO, StringDB, and TissueAtlas.

The systematic review was carried out according to the Preferred Reporting Items for Systematic Review and Meta-Analysis (PRISMA 2020). A flow chart is provided in Figure 1.

## 3. Results

### 3.1. Problems of Research of Surgery Outcomes

MicroRNA analysis can lead to different data in independent studies. This fact can be explained by several reasons: differences in the technical approach (cohort, sample), differences in the choice of sample for research (blood serum, plasma, hippocampal tissue), differences in the type of cells selected for research (neurons, astrocytes, oligodendrocytes, microglia), the choice of the studied organism (patient, experimental model), the difference in the type of epilepsy, as well as the difference in time elapsed after the seizure [26,27,28].

In a study on miRNOME profiling of human hippocampal tissue in patients with TLE with hippocampal sclerosis, microRNAs, which were extensively studied in epilepsy, did not show aberrant expression (miR-146a, miR-132-3p, miR-132-5p, miR-134-5p). This suggests a unique expression profile for TLE patients with hippocampal sclerosis [26].

A study conducted by Benedittis et al. shows a large number of correlations [28]. Most importantly, correlation patterns differ between drug-resistant and drug-responsive patient groups. In the group of drug-resistant patients, associations were found between miR-146a and gender and miR-223 and age of onset of the disease. A group of drug-sensitive patients showed co-expression of miR-142 and miR-223, as well as miR-146a and miR-132. In the overall analysis of both cohorts, miR-142 and miR-223 were associated with drug resistance.

The difference in microRNA expression also depends on the time of blood sampling. A study conducted by Surges et al. showed differential microRNA expression in serum that was statistically significant only at the nearest postictal point in time (30 min after the seizure) [29]. This result was observed in 215 different microRNAs, while at later time intervals (3–6 h, 20–28 h, 3–6 days), there was no difference in expression compared with the control. However, in four patients who had seizures during sleep, aberrant expression was also observed at later postictal intervals. This indicates a correlation between the aberrant expression and the seizure. However, this correlation is not universal; microRNA expression in some patients can also be observed at later postictal intervals. MiR-663b begins to show overexpression in the first 30 min, and it only increases in subsequent periods of time. There is also a correlation between the total seizure duration and the relative change in microRNA expression levels: with an increase in the total seizure time, the relative expression level decreases. Similar data were obtained in the study by Sun et al. [30]. When analyzing the expression of microRNA in a blood serum, a difference in expression was found at the very beginning of the seizure, which the authors characterized as the presence of signs on the EEG and in the post-seizure state, when no signs were observed on the EEG. An increase in the expression of miR-30a, miR-378, miR-106b, and miR-15a at the beginning of a seizure, as well as its decrease in the postictal period, was reliably confirmed.

In experimental models, the dependence of the expression profiles of specific microRNAs and the stage of the disease course was established. Thus, the greatest amount of miRNAs are expressed in the acute stage of the disease: 296 microRNAs and 169 microRNAs are expressed in the chronic stage, and 71 in the latent phase. A relationship was also found between the stage of the disease and a set of specific microRNAs. Thus, 174 microRNAs are specific for the acute phase, 53 for the chronic phase, and 17 for the latent phase. The number of microRNAs found in all stages of the disease was 24 microRNAs [31].

A study of Baloun et al. showed a relationship with the age of onset of the disease and microRNA expression profiles [32]. A total of 123 differentially expressed microRNAs were found in patients with the age of the onset up to 10 years, 130 microRNAs for the group between 10 and 19 years old, and 80 microRNAs for the group over 20 years old, while 49 microRNAs were detected for the total age group with aberrant expression.

Difficulties in the analysis of publications and the design of subsequent studies arose at the stage of selecting samples for research. The study of circulating microRNAs faces the problem of the selection of biological material: blood, blood serum, blood plasma, or cerebrospinal fluid. The cerebrospinal fluid has advantages due to contact with the extracellular space of the pathological area [33]. Blood serum and blood are characterized by high microRNA stability [30,34]. Many studies do not explain the choice of a specific biological fluid [29,35,36,37,38]. However, the exact answer to the question of how the specific liquid will affect the microRNA expression profiles has not yet been given [29]. A problem in the study of brain tissue is the discrimination of patients by groups with or without hippocampal sclerosis, which affects microRNA expression profiles [39,40]. Hippocampal sclerosis leads to loss of neurons and gliosis in hippocampal regions, which also forces us to pay attention to expression profiles in different regions when examining hippocampal tissue [40,41,42].

Expression of microRNA depends on the type of sample being studied. MicroRNAs, showing significant expression in the discrimination of healthy patients and patients with temporal lobe epilepsy, and miRNAs isolated from the blood, in terms of their expression level, are not equal to nucleic acids isolated from brain tissues [43].

Using an experimental model in rats, it was found that microRNAs can be highly specific for certain cell types. The study compared four types of cells: neurons, astrocytes, oligodendrocytes, and microglia. For 116 out of 351 studied microRNAs, it was found that it is expressed specifically with a difference of more than five times when comparing one cell type with the average value for the other three [27].

A problem in such studies is the inadequacy of the sample, which can be solved by creating an animal model of epilepsy. Most often, rodents are used for such models, but such studies cannot fully reflect all the patterns of epileptogenesis. However, microRNAs are often conserved and similar in humans, mice, and rats; for example, hsa-miR-155 differs by only one nucleotide from rodent miR-155. This suggests that many targets of posttranscriptional silencing are retained in the species described above, which makes it possible to study the functions of microRNAs and physiological pathways in a rodent model [44].

### 3.2. Biomarkers of Response to Surgical Treatment: Conducted Researches 

Currently, the search for microRNA-based biomarkers of response to surgical treatment of TLE continues. Thus, three microRNAs were studied: miR-27a-3p, miR-328-3p, and miR-654-3p [3]. The choice of microRNA data was justified by data obtained in another, larger study conducted by Raoof et al., looking for a epilepsy biomarkers [36]. The significance of the selected microRNAs was determined both as diagnostic biomarkers and prognostic biomarkers of the outcome of surgical treatment. Sera were analyzed from 28 patients, 14 of whom had a good surgical outcome (Engel I) and 14 with a poor surgical outcome (Engel III–IV), compared with 11 healthy volunteers. The analysis was carried out by the ROC-curve method. As a result, miR-27a-3p did not show any significant discriminatory result, despite previous opposite data; however, the authors attribute this fact to a difference in the choice of blood fraction: serum or plasma. MiR-328-3p showed significant discriminatory ability as a diagnostic biomarker for TLE (AUC was 93.5%), but sufficient power in the discrimination of Engel I and Engel III-IV patient groups was not found. MiR-654-3p performed the best result as both a diagnostic and prognostic biomarker (AUC = 74.7% and AUC = 73.6%, respectively). Thus, miR-654-3p is a potential prognostic biomarker of the surgical outcome of epilepsy. Bioinformatic analysis demonstrates many pathways that are often associated with epilepsy: ATP-binding cassette drug transporters, glutamate transporter SLC7A11, and TP53. A specific target for miR-654-3p is GLRA2, the glycine receptor subunit. Data about pathways and gene-targets are compiled in Table 1.

The co-expressed GABAergic receptor-targeting miR-629-3p, miR-1202, and miR-1225-5p were analyzed for potential significance in predicting surgical outcome in the treatment of epilepsy based on unpublished microarray analysis by the same group of scientists. Expression analysis was carried out for both circulating microRNAs as well as for tissues microRNAs (hippocampus and amygdala). No significant result was obtained as a biomarker of prognosis; however, these microRNAs showed good results in the discrimination between groups of healthy people and patients with temporal lobe epilepsy with hippocampal sclerosis. MiR-629-3p analysis showed increased expression in the blood of people with epilepsy compared with controls, but did not reveal a difference between the Engel I and Engel III–IV groups. There was also no significant difference in expression in either the hippocampus or the amygdala between control and surgical outcome groups. MiR-1202 also revealed differences in blood expression between controls and patients with temporal lobe epilepsy, but did not result in discrimination against the Engel groups. However, when analyzing the expression in the amygdala, there was a decrease in expression in only the Engel III–IV group compared with the control. MiR-1225-5p was also more expressed in patients with temporal lobe epilepsy compared with the control in the blood, but the significance in distinguishing Engel groups was not revealed [43]. 

There are data on aberrant expression of some microRNAs in the post-resection period. As shown in the study by Huang et al., in patients with overexpression of miR-155-5p in the post-resection period, the tendency of recurrent seizures is increased [44]. The choice of this microRNA was due to its participation in inflammatory processes. This microRNA promotes the release of TNF-α mediators and regulates by the p53 factor; therefore, it is considered a pro-inflammatory and pro-apoptotic microRNA. The analysis performed showed the significance of miR-155-5p in the processes of oxidative stress by affecting the transcript of the Sesn3 gene. Suppression of miR-155-5p expression inhibits apoptotic processes and reduces neuronal loss. Participation in inflammatory processes is characterized by its promotion of the release of TNF-α mediators and its regulation by the p53 factor; therefore, it is considered a pro-inflammatory and proapoptotic microRNA, which could be a factor that caused recurrent seizures. The analysis performed showed the significance of miR-155-5p in the processes of oxidative stress by affecting the transcript of the Sesn3 gene. Suppression of miR-155-5p expression makes it possible to restore translation of the Sesn3 protein, which inhibits apoptotic processes, oxidative stress, and reduces neuronal loss.

## 4. Discussions

Based on the above, it is possible to set a framework for a potential prognostic microRNA-based biomarker. First of all, it is necessary to determine the nature of the relapse. Early relapses are caused by a distant focus of the primary disease, as a result of which biomarkers of epilepsy itself are the goal of the search. Biomarkers of epileptogenesis will be the target for searches in late relapse, as a new epileptic focus develops.

Thus, miR-134 requires special attention. The connection of miR-134 with epileptogenesis is proved by the data previously obtained, according to which miR-134 expression was not manifested during short-term generalized seizures [45]. This indicates that under physiological conditions miR-134 upregulation is not only a response to increased neuronal activity, but it can also directly affect epileptogenic and pathogenic brain activity.

MiR-30a is also proposed as a marker of epilepsy. At the onset of seizures, miR-30a expression level was positively associated with the frequency of seizures. Sun et al. suggest a correlation between the inhibition of CAMK4 gene products, which has not been previously studied in epilepsy, and the miR-30a pathway, and consider this microRNA as potentially useful for predicting conditions after a seizure in further study [30]. 

At the moment, there are a number of studies investigating the search for biomarkers of temporal lobe epilepsy. Information about the obtained data is compiled in Table 2.

Statistically significant expression was observed only in the postictal period (30 min after the seizure) in 215 microRNAs, which indicated their association with seizures, which was the factor in including these microRNAs in the analysis. Four patients underwent a more precise study, in which increased microRNA expression was observed at later time intervals, which was associated with the onset of bilateral seizures in sleep. MiR-663b had an aberrant expression profile, increasing in the first 30 min and reaching a peak at 20–28 h. Expression of miR-143-3p and -145-5p was inversely proportional to the total duration of the seizure [29].

The miRs-143-3p/145-3p are involved in vascular stabilization and smooth muscle contractility, and serve as a common biomarker of disease activity in psoriasis [49,50]. MiR-365a-3p and miR-532-5p are involved in the regulation of various types of cancer [51,52]. MiR-663b has been identified as a circulating biomarker in various diseases such as bladder cancer, myocardial infarction, and T-cell lymphoma. It is also highly expressed in brain tissues [53,54].

Microarray analysis showed significant expression of 50 microRNAs. The selected microRNAs are involved in such biological processes as homophilic cell adhesion, synaptic transmission, signal transduction, cell adhesion, downregulation of transcription from RNA polymer, positive regulation of transcription from RNA polymer, protein phosphorylation, and apoptotic processes. Significant signaling pathways have been identified: axon guidance, pathways in cancer, regulation of the actin cytoskeleton, focal adhesion, calcium signaling pathway, MAPK signaling pathway, and PI3K-Akt signaling pathway [38].

ROC analysis served as the basis for suggesting some microRNAs as diagnostic biomarkers of epilepsy, as a result of which six microRNAs turned out to be significant: miR-3613-5p, -4668-5p, -4322, -8071, -197-5p, and -6781-5p. Among them, miR-8071 showed a correlation with the duration of the disease and the frequency of seizures, on the basis of which it is concluded that exosomal microRNAs are actively involved in epileptogenesis [38].

The choice of microRNA was based on a previous study that showed the relationship of miR-134-5p with epilepsy, which is specific to brain tissue and is involved in the control of the morphology of dendritic spines of pyramidal neurons in the CA3 region of the hippocampus [45]. 

The ability of miR-134-5p to discriminate between healthy people and patients with mTLE was assessed using ROC analysis, which showed an area under the curve = 0.75, a sensitivity of 65%, and a specificity of 75%, and no significance was observed in the discrimination of patients diagnosed with focal cortical dysplasia type one and the control group. In addition, it has been shown that there is no difference in expression profiles between the drug-resistant and drug-sensitive group [46].

Four microRNAs (miR-145, miR-181c, miR-199a, and miR-1183) were selected based on microarray analysis of blood and hippocampal tissue samples, which showed co-expressed microRNAs and subsequent bioinformatic analysis, data not specified by the authors. No significant result was found between the Engel groups [47].

A study of Antônio et al. showed that miR-145 is overexpressed in both hippocampal tissue and human blood samples, and its expression is also upregulated in mouse hippocampal tissue after epileptic seizures [47]. In addition, miR-145 showed reduced expression in patients with intracranial aneurysms; this expression was associated with phagocyte migration, movement and proliferation of mononuclear leukocytes, T-lymphocyte stimulation, and macrophage differentiation. Since epilepsy is currently associated with the immune system, this relationship is significant in understanding epileptogenesis.

For the selection of microRNAs after sequencing, the authors used the following criteria: (1) at least 10 copies in each group and (2) fold change (|log2 patient/control|) > one between groups (*p* < 0.05) [27]. Accordingly, 42 microRNAs were selected and validated by qRT-PCR. An assessment of the significance of discrimination between mTLE + HS patients from mTLE-HS was carried out by ROC analysis and revealed the following areas under the curve: for hsa-miR-129-5p, -214-3p, -219a-5p, -34c-5p, -421, and -184 were 0.735; 0.894; 0.708; 0.764; 0.873; and 0.923, respectively. MiR-184 showed the best result with AUC = 0.923, sensitivity 88.9%, and specificity 83.3%. It was also shown that a decrease in miR-184 expression correlates with an increase in the frequency of preoperative seizures, as a result of which the authors conclude that this microRNA plays an active role in the development of mTLE + HS [35].

Bioinformatic analysis of microRNA data provided information on the following biological pathways: hippo, p53, TGF-β, HIF-1, mTOR, and neurotrophin, of which the Hippo signaling pathway, p53 signaling pathway, TGF-β signaling pathway, HIF-1 signaling pathway, and mTOR signaling pathways are associated with neuronal function and the development of epilepsy, as well as cancer pathways of glioma, myeloid leukemia, prostate, melanoma, bladder, renal leukemia, colorectal, and pulmonary [35].

The analysis carried out by the authors indicates that the microRNA complex (miR-19b-3p, -21-5p, -451a) can be used to discriminate the mTLE group from the control, and the complex of miR-21-5p and miR-451a discriminates against patients with epileptic status from mTLE, control, and other neurological diseases [33].

MiR-106b showed itself as the best biomarker. The researchers, as a result of the bioinformatics analysis, note that microRNAs can participate in epileptogenesis through the regulation of inflammation and apoptosis [33].

In our opinion, the problem of searching for potential biomarkers of surgical outcome lies in the insufficient binding of the selected microRNAs to any biologically significant criterion. Thus, in the case of the team of E. S. Ioriatti et al., the choice of microRNAs for analysis was based on the results of a previous study [36], which, in turn, was based on a two-stage study design that included a miRNOME review phase and a validation phase; the results of mathematical analysis on the significance of differential expression; analysis of tissue microRNA expression by reviewing a tissue atlas (link) and in situ hybridization; and bioinformatics path analysis [3]. However, microRNAs obtained in this way (miR-27a-3p, miR-328-3p, miR-654-3p) do not reflect the dependence of microRNA expression profiles on the time elapsed after the seizure. Blood for analysis was taken 24 h after the seizure, because, according to the authors, this allowed them to register any potential mechanisms for the prolonged release of microRNAs into the bloodstream, while avoiding clinical factors that could lead to bias. 

These data, however, disagree with the results obtained in the study of Surges et al. [30], who noticed a significant differential microRNA expression only in the postictal period (the first 30 min after the seizure), which was also transient in its manifestation [22]. It is important to note that in the study on the outcome of surgical treatment, blood sampling was carried out in a standardized manner upon admission to the operating room, which is of fundamental importance since the time elapsed since the last attack, as well as the effect of anticonvulsants before the operation, are not taken into account. Passive and active mechanisms of microRNA release from tissues affect their expression profile [33]. As a result, microRNA expression can be affected by both external factors and pathophysiological processes, where expression aberration in the latter case can serve as a characteristic of a pathological condition [55].

The question of the timing of sampling of autopsy material also remains open, although there are data in experiments on rats that microRNAs are stable in dead tissues and their expression profile remains unchanged for 32 h [26].

A large number of studies of drug-resistant temporal lobe epilepsy indicate the course of antiepileptic drugs treatment. It is still not known whether they have any effect on the microRNA expression profile. There are data on the absence of such correlations, but other studies show that microRNAs can increase sensitivity to antiepileptic drugs [36,46,56].

It is also known that expression patterns of some microRNAs, such as miR-886-3p, can correlate with human age [33]. However, this fact was not further studied, since the correlation was revealed in the process of checking the effect of this factor on expression, as a result of which miR-886-3p was excluded from further analysis.

A study by Gattás et al. was based on microarray analysis of microRNAs of hippocampal tissue and blood samples, which resulted in the selection of co-expressed microRNAs in both variants (co-expression was based on the results of mathematical analysis) [43]. In this case, many criteria for selecting microRNAs for analysis remain unaccounted for, potentially having an effect in the selection for significance in epilepsy. Thus, the expression of miR-629-3p (Figure 2), although noted in the brain tissue, is expressed nonspecifically and is noted in all body systems. A similar situation is seen in miR-1202 (Figure 3), which, among other things, lacks experimental data on expression. The most favorable situation in terms of selection for analysis for a potential biomarker is observed in miR-1225-5p (Figure 4 and Figure 5), the expression of which is most specific for brain tissue.

A recent review article by Martinez et al., aimed at analyzing data from studies on the study of microRNAs as biomarkers of TLE, highlights the following potential markers: miR-129-2-3p, miR-142-5p, miR-145- 3p, miR-153, miR-199a-3p, and miR-339-5p [57].

The selected experimental miRNA data pathways using the “MiRTarBase” database (targets presented in the literature more than two times and/or with at least one confirmed strong evidence were selected) and the analysis of the pathways using the ShinyGO v.0.77 and StringDB software indicate direct participation of these miRNAs in epileptic processes (Figure 6, Figure 7 and Figure 8).

Thus, we see that microRNAs, which are often described in publications and proposed as biomarkers, are also involved in processes characteristic of epilepsy. The most significant biological process in both analyses is any involvement in oxidative stress, which a large body of literature indicates correlates with epilepsy and may also be associated with drug resistance [53,58].

One of the most important functions is cell apoptosis associated with cell death after an attack, which is represented by several biological processes. The implementation of the apoptotic pathway can be represented by internal (mitochondrial) and external (death receptors) causes [59].

Speaking on international affairs, the importance of mitochondrial functions is emphasized, which directly control apoptosis and cell necrosis, adenosine triphosphate production, fatty acid oxidation, regulation of the amino acid cycle, synthesis of neurotransmitters, and regulation of calcium ion transport into the cytosol [60]. The influence of mitochondria on epileptic processes is reflected in the disruption of their functions, and the dysfunction of mitochondria is extremely susceptible to oxidative stress, which is expressed in the previous results in the greatest significance of oxidative stress in all pathways of target genes potentially suitable for use as microRNA biomarkers.

The Bcl-2 protein family positively or negatively regulates apoptosis. The internal pathway of apoptosis is regulated by cytochrome from mitochondria, which activates a cascade of reactions leading to an increase in the concentration of calcium ions in cells, activation of the proapoptotic protein Bcl-2, or reactive oxygen species [61].

The extrinsic pathway of apoptosis is initiated upon activation of death receptors on the cell surface. The genes regulated by these microRNAs are responsible for the regulation of the binding of the BH3 domain, which is a cell death agonist, while interacting with the BAX protein.

There are numerous studies based on animal models. The study of epilepsy in animal models alone has several disadvantages. Firstly, it does not exactly repeat the disease under study, since the biochemical processes of experimental animals and humans are different, as a result of which the results require careful interpretation in the context of a human disease. Secondly, there are fundamental differences in the approaches to experimental models themselves, since each implements different methods that can cause epileptic seizures, which can lead to ambiguous results and difficulties in comparing data. Processes associated with epileptogenesis may be model-specific and not representative of epilepsy as a whole [32].

The reason for the changes in microRNA expression profiles in patients with TLE is still unknown. This can be caused directly by changes in brain tissue (inflammatory and neurodegenerative processes) or by seizures. Further studies are needed to identify these mechanisms; however, the data already available confirm the relationship between seizures and changes in microRNA expression [62]. The largest number of aberrantly expressed miRNAs is observed in the first 30 min after the seizure; however, the authors note that the reason for such changes in miRNA expression is unknown, since activity in muscle cells during seizures could be reflected. The same study showed that some microRNAs can increase their activity not only in the first 30 min after a seizure, but also in the subsequent 3–6 and 20–28 h; however, such changes were noted only in patients who experienced seizures during sleep. Biological pathways have not been analyzed, which makes it impossible to answer the true reason for these results. 

However, the availability of such data suggests that those microRNAs that can act as a biomarker of surgical outcome may be dependent on the time elapsed since the attack. However, for unambiguous conclusions, additional studies of this phenomenon are needed, including a bioinformatic analysis of the pathways characteristic of miRNAs of each time interval.

MicroRNAs are often tissue-specific molecules, since their function is post-translational silencing of those genes that are characteristic of a particular area. Since the brain tissue is heterogeneous, there is specificity in miRNA expression for certain regions. This is the hippocampus, which is key in mesial temporal lobe epilepsy and often concomitant hippocampal sclerosis. The hippocampus is represented by four regions (CA1, CA2, CA3, and CA4) [63]. Thus, it is known about the relationship of miR-134 and the morphology of the spines of pyramidal neurons. Inhibition of miR-134 led to a decrease in the density of pyramidal neuron spines in the CA1 and CA3 regions, which had an anticonvulsant effect. Our assumption is that the miRNA selected for analysis should at least be expressed in the tissue that is involved in the pathogenesis of the disease.

The relevance of this study, in our opinion, lies in the description of some important factors that limit the choice of miRs to study them as prognostic biomarkers. This systematic review was limited to mesial temporal lobe epilepsy, which could include hippocampal sclerosis. Other types of epilepsy were not included to establish a greater focus on specific mechanisms of pathogenesis. Additionally, childhood forms of mesial temporal lobe epilepsy were not considered in this study.

## 5. Conclusions

The study of microRNA expression profiles as prognostic biomarkers of the outcome of surgical treatment of temporal lobe epilepsy is an acute and urgent task. Further data accumulation in this area should continue.

A good approach in this area is to compare miRNA expression profiles in temporal lobe epilepsy and in status epilepticus [29]. Such an approach makes it possible to classify microRNAs according to their functions in epileptic processes, and also helps to clarify the potential ability of a particular microRNA to be a marker of a certain pathology.

We suggest that it is important to take into account a number of factors, such as: (a) conditions for the collection of initial blood samples; (b) timing of collection of post-seizure blood samples; (c) tissue specificity of the miRNA under study; and (d) the type and duration of therapy, taking into account specific antiepileptic drugs.

## Figures and Tables

**Figure 1 ijms-24-05694-f001:**
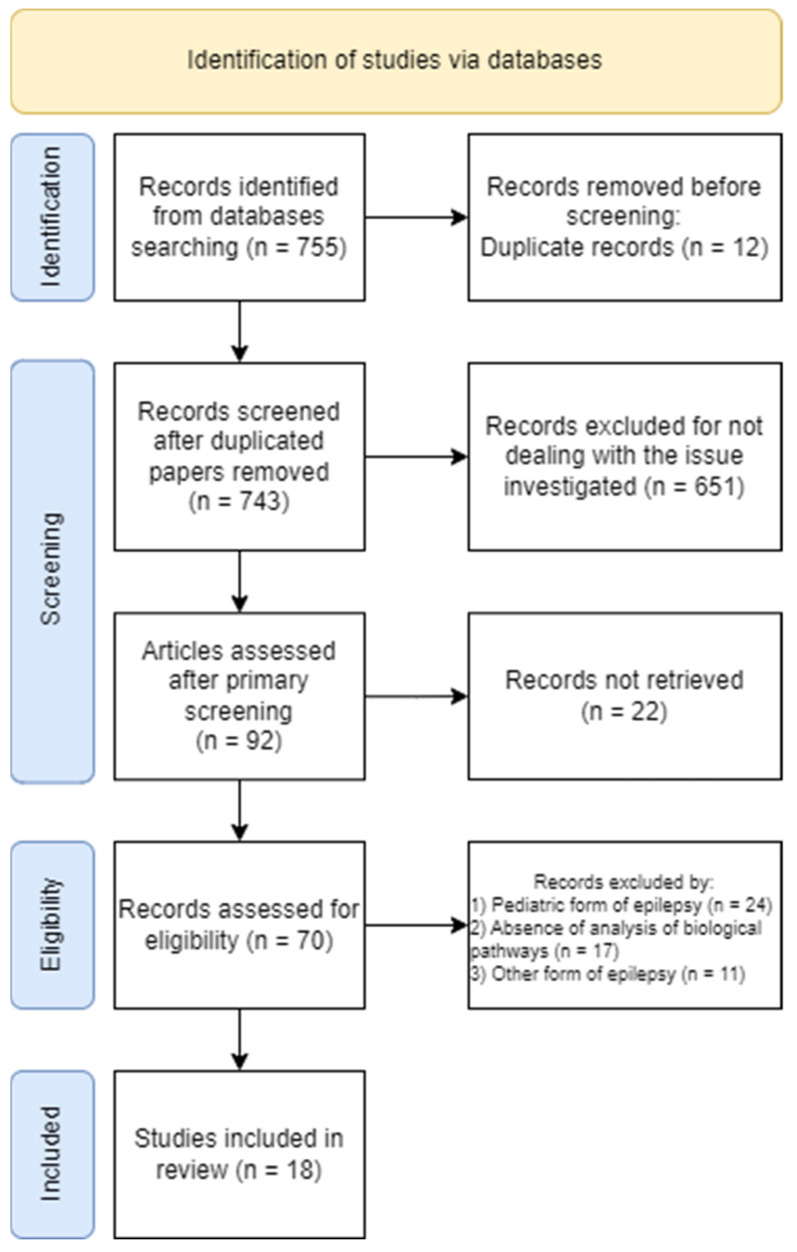
Flow chart diagram visualizing the database searches, number of publications identified, screened, and final full texts included in the present systematic review.

**Figure 2 ijms-24-05694-f002:**
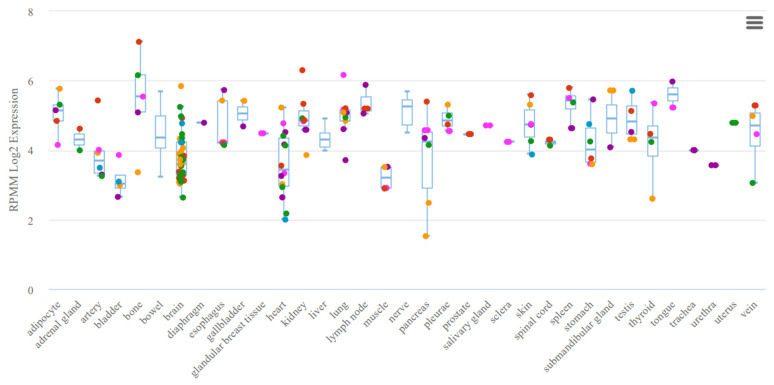
Tissue expression of miR-629-3p.

**Figure 3 ijms-24-05694-f003:**
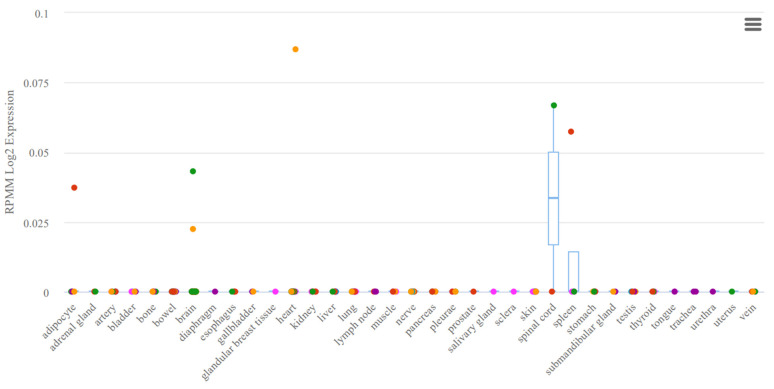
Tissue expression of miR-1202.

**Figure 4 ijms-24-05694-f004:**
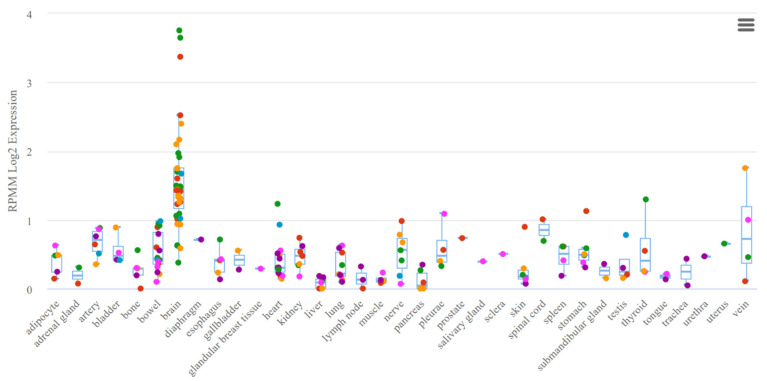
Tissue expression of miR-1225-5p.

**Figure 5 ijms-24-05694-f005:**
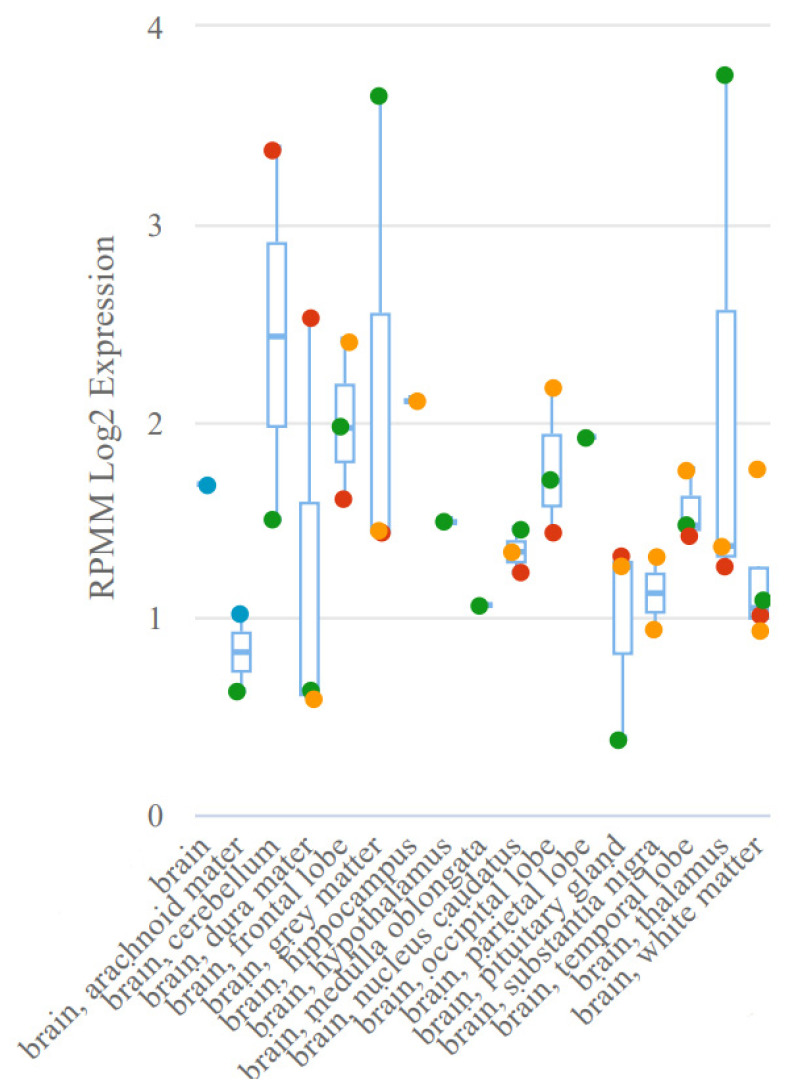
Expression of miR-1225-5p in brain tissue.

**Figure 6 ijms-24-05694-f006:**
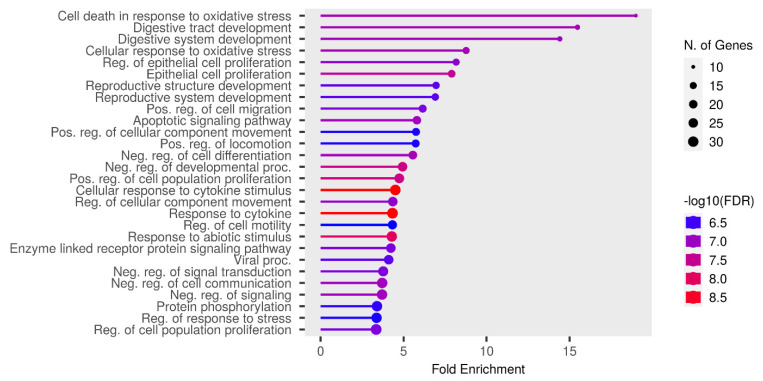
Plot of the most significant pathways in which the target genes of the selected miRNAs are involved.

**Figure 7 ijms-24-05694-f007:**
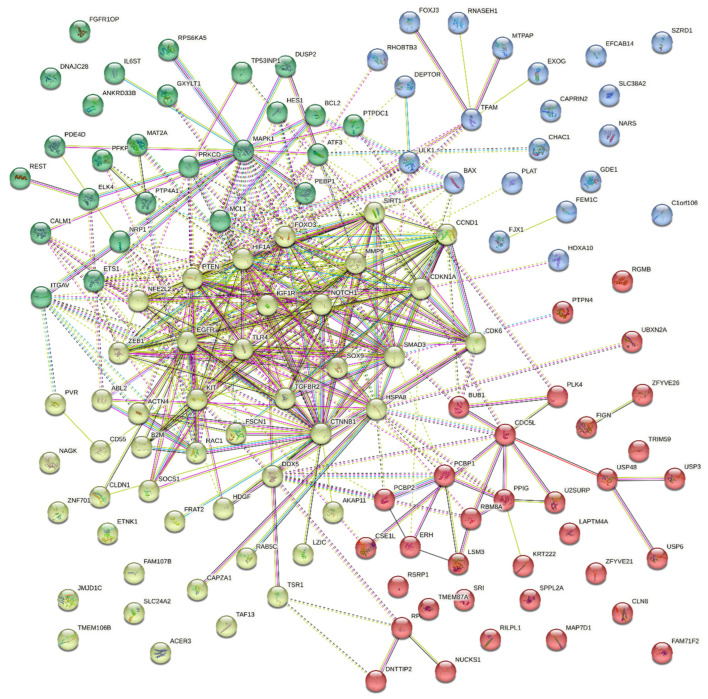
Network of all target genes.

**Figure 8 ijms-24-05694-f008:**
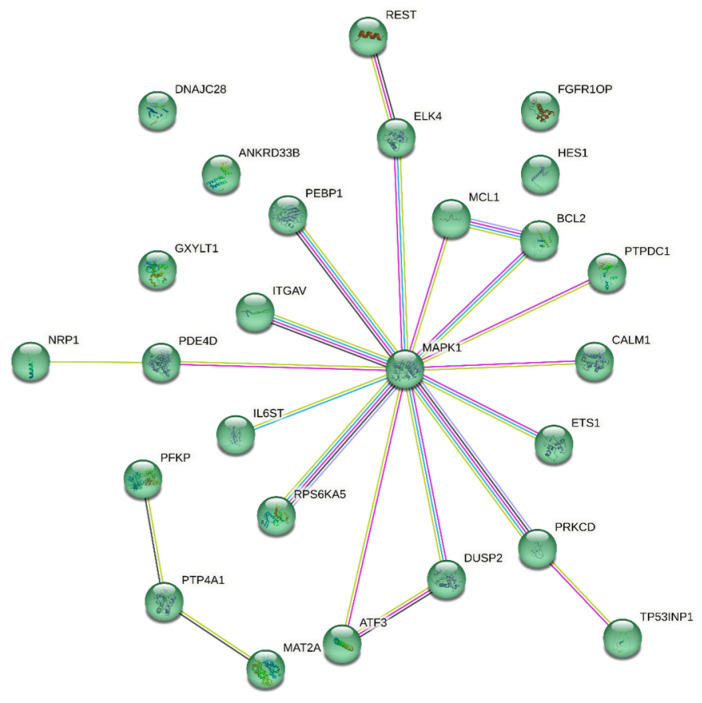
Cluster of target genes most characteristic of epileptic processes.

**Table 1 ijms-24-05694-t001:** Associations and significant gene targets of potential prognostic miRNAs.

microRNAs	Associations	Genes and Biological Pathways	Authors
miR-654-3p	Prognostic biomarker of surgical outcome Diagnostic biomarker of TLE	*ABCA1*, *ABCG2*, *APC*, *BACE1*, *CCND1*, *CD44*, *CDKN1A*, *CNN3*, *COG7*, *DPYD*, *EGFR*, *FOXO1*, *GLRA2*, *GPD2*, *GRB2*, *IGF-1*, *KRAS*, *MAP2K4*, *MET*, *MYT1*, *NOVA1*, *PIK3CG*, *PLK2*, *PPARG*, *PSAP*, *SLC16A1*, *SLC6A1*, *SLC7A11*, *SP1*, *ST6GALNAC3*, *THRB*, *TP53*, *YWHAZ*	Ioriatti et al. [3]; Bioinformatics analysis by Raoof et al. [29]
miR-328-3p	Diagnostic biomarker of TLE		
miR-27a-3p	Has no discrimination significance		
miR-629-3p	Diagnostic biomarker of TLE	GABAergic receptors genes	Gattas et al. [43]
miR-1202	Diagnostic biomarker of TLE		
miR-1225-5p	Diagnostic biomarker of TLE		
miR-155-5p	Participation in recurrent seizures	*TNF-α*, p53, *SESN3*; Inflammatory processes, apoptotic processes, oxidative stress	Huang et al. [44]

**Table 2 ijms-24-05694-t002:** miRs as biomarkers of TLE.

Disease	Cohorts	Sample Material	Expression Profile		Link
Drug resistant mTLE + HS evolving to a bilateral convulsive seizure	*Inclusion criteria:* (a) adults (b) MRI+; (c) cranial MRI- for other pathologies *Experimental group*: *n* = 15 people; mean age = 43 years; mean onset age = 15.6 years; mean disease duration = 27.4 years; 6 men, 9 women	Serum (1 before seizure 2 within 30 min after seizure 3 within 3–6 h after seizure 4 within 20–28 h after seizure 5 within 5–6 d after seizure)	Upregulated	miR-143, miR-145, miR-532, miR-365a	Surges et al. [29]
Drug resistant mTLE + HS	*Control group*: *n* = 40 people; mean age = 28.72; 22 men, 18 women *Experimental group*: mTLE + HS: *n* = 40 people; mean age = 27.56 years; mean onset age = 12.45 years; mean disease duration = 14.52 years; 25 men, 15 women	Blood plasma	Downregulated	miR-3613-5p, miR-4668-5p, miR-4322, miR-8071, miR-197-5p, miR-6781-5p	Yan et al. [38]
mTLE + HS, MRI+-review cohort Drug-resistant mTLE + HS/-HS and drug-sensitive mTLE + HS/-HS, MRI+–validation cohort	*Control group: n* = 16 people; 6 men, 10 women *Experimental group:* mTLE + HS: *n* = 14; onset in the first 10 years of life = 6 people; onset of the disease later than 10 years of life = 8 people; 6 men, 8 women *Validation cohort:* *Control group: n* = 83 people; 35 men, 48 women DR mTLE + HS: *n* = 38 people; onset in the first 10 years of life = 14 people; onset after 10 years of life = 24 people; 16 men, 22 women DS mTLE-HS: *n* = 27 people; onset in the first 10 years of life = 11 people: onset after 10 years of life = 16 people; 12 men, 15 women	Blood plasma	Downregulated	miR-134-5p	Avansini et al. [46]
mTLE + HS	*Control group: n* = 9 people; no neurological or psychiatric history; autopsy material taken within 15 h of death *Experimental group:* mTLE + HS: *n* = 20 people; *n* of Engel I = 10 people; mean age = 35.2 years; 3 men, 7 women; *n* of Engel III–IV = 10 people; mean age = 38.2 years; 3 men, 7 women	Hippocampal tissues	Downregulated	miR-145	Antônio et al. [47]
mTLE + HS	*Control group: n* = 10 people *Experimental group:* mTLE + HS: *n* = 20 people; *n* of Engel I = 10 people; mean age = 35.2 years; 3 men, 7 women; *n* of Engel III–IV = 10 people; mean age = 38.2 years; 3 men, 7 women	Blood	Upregulated	miR-145, miR-181c, miR-199a, miR-1183	Antônio et al. [47]
mTLE + HS; mTLE-HS	*Review cohort* Control/mTLE-HS: *n* = 6 people; mean age = 22 years; mean onset age = 19 years; mean disease duration = 6.5; 3 men, 3 women mTLE + HS: *n* = 6 people; mean age = 29 years; mean onset age = 19.5 years; mean disease duration = 12.17; 2 men, 4 women *Validation cohort* Control/mTLE-HS: *n* = 18 people; mean age = 23 years; mean onset age = 11 years; mean disease duration = 12.44; 7 men, 11 women mTLE + HS: *n* = 18 people; mean age = 27 years; mean onset age = 9.5 years; mean disease duration = 16.44; 6 men, 12 women	Blood plasma	Upregulated	miR-129-5p, miR-214-3p, miR-219a-5p, miR-34c-5p	Huang et al. [35]
			Downregulated	miR-421, miR-184	
mTLE versus status epilepticus and other neurological diseases	*Control group: n* = 28 people; 10 men, mean age = 29.8 years; 18 women, mean age = 39.9 years *Focal status epilepticus: n* = 10 people; 4 men, mean age = 72.2 years; 6 women, mean age = 72.8 years *Generalized tonic-clonic seizures: n* = 3; 3 men, mean age = 76 years *Non-convulsive SE: n* = 5 people; 3 men, mean age = 68 years; 2 women, mean age = 63.5 years *Other neurological diseases:* Alzheimer’s disease: *n* = 9 people; 6 men, mean age = 73.3 years; 3 women, mean age = 72 years Multiple sclerosis: *n* = 10 people; 2 men, mean age = 62 years; 8 women, mean age = 32.3 years Other: *n* = 6 people; 3 men, mean age = 74.61 years; 3 women, mean age = 60.6 years	Cerebrospinal fluid	Downregulated	miR-19b	Raoof et al. [33]
Epilepsy of various etiologies and types of seizures	*Control group: n* = 112 people; mean age = 31.8 years; 53 men, 59 women *Experimental group*: *n* = 117 people; mean age = 29.8 years; mean disease duration = 6 years; 60 men, 57 women	Serum	Upregulated	let-7d-5p, miR-106b-5p, miR-130a-3p, miR-146a-5p	Wang et al. [48]
			Downregulated	miR-15a-5p, miR-194-5p	

mTLE-mesial temporal lobe epilepsy. HS-hippocampal sclerosis. MRI+-MRI-positive. SE-status epilepticus. *N*-amount of patients. DR-drug resistant. DS-drug sensitive.

## Data Availability

Not applicable.

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
