# Peer review of "MicroRNAs as Biomarkers of Surgical Outcome in Mesial Temporal Lobe Epilepsy: A Systematic Review"

_ijms, 2023, doi:10.3390/ijms24065694_

Round 1
Reviewer 1 Report
Intresting review with author opinions and inspiration.
A professional English edit is necessary to improve the paper's structure, conncection, discussion, and reference, for example, the structural line 40 can be better summarised. Similar to other opening sentences, could be improved.
Avoid using word to word copying in abstract into main text. For instance, line 12 verse 31 and line 15 verses 57. Rest of the abstract to follow.
gramma:
'Speaking on international affairs', 365.
"there are numerous studies based on animal models" in line 381. Need re-structure
"A good help" 391. etc
Accept after professional English edit.
Below are some specific comments:
1 It would be nice to provide more information on the associations found between miRNAs and variables indicating surgical treatment results in patients with drug-resistant and drug-responsive conditions?
2 How does the timing of blood collection impact the expression of miRNA in TLE patients, and explain the significance of this?
3 What specific challenges are associated with analyzing miRNA expression profiles in different brain tissues, especially those related to hippocampal sclerosis?
4 Please ellaborate in more detail how miR-27a-3p, miR-328-3p, and miR-654-3p affect the likelihood of surgical treatment success for TLE?
5 What were the specific findings of the analysis of miRNAs 629-3p, 1202, and 1225-5p as potential indicators of the success of epilepsy surgery?
6 How does miR-155-5p play a role in the recovery process following TLE surgery, and what is the reason for this?
7 Can you provide more explaination on the correlation between miRNA-886-3p and human age, and what this means for understanding the role of miRNAs in aging?''
Author Response
Dear Reviewer,
Thank you very much for your interest in our manuscript and comments, we really appreciate it. We have tried to answer your remarks and made clarifications in manuscript.
We tried to improve the sentence structure in the text. Abstract has been modified and rewritten. All changes are highlighted in yellow.
- Unfortunately, there is no more detailed information on this issue. This is all the data that was provided in the study we link to. Moreover, due to the fact that we are talking about the outcome of surgical treatment, which, as a rule, is not indicated for drug-sensitive patients, we do not consider such an association.
- In the discussion section, we outlined why the timing of blood sampling is important.
- Information on this issue has been added to the discussion section.
- , 5), 6) A more detailed description of the biological functions of these miRNAs has been added to the Results section.
7) This is all the available information provided in the study we link to. In addition, the presence of this correlation was a factor for excluding this miRNA from the study. Unfortunately, we cannot provide more data on the role of this miRNA in aging, but this question was not the aim of our study.
We tried to take into account all your comments.
Sincerely,
Authors
Reviewer 2 Report
Dear authors, it is a great focus on the topic, but I would like to tell you that this review needs editing before it can be considered, please see the following list of points that need to be considered:
Question 1: The introduction is very short with a lot of information included making it difficult to read. I would suggest the authors increase the size making it easier to understand.
Question 2: In the results section, the authors include several comments trying to explain the results, even if such comments help to understand the result, the integration of these comments together with the material and method information in the results section makes it very complicated and redundant. I would suggest the author concentrate the commentary on the findings in the conclusions section.
Question 3: In the conclusion section, there are many long sentences that make the reader lose track. I would suggest the author to design a scheme that organizes and sorts his comments on the results in a simpler and more understandable way.
Overall excellent statistical analysis, supported by explanatory box plots and tree graphs and recent references.
My best regards
Author Response
Dear Reviewer,
Thank you very much for your interest in our manuscript and comments, we really appreciate it. We have tried to answer your remarks and made clarifications in manuscript. All changes are highlighted in yellow.
Мы учли все все ваши комментарии и сделали уточнения в рукописи
- The introduction has been updated and modified.
- We have made corrections according to your comments.
- We have modified the text of the manuscript.
We tried to take into account all your comments.
Sincerely,
Authors
Reviewer 3 Report
All references go to the end of the phrase.
In the studies taken over by XnameX et al. just pass the year, it will be in the references.
Lines 105-106 – EEG signs, not symptoms
Subchapter 3.2 – at its end, a table (2x2)/schema can be added to include the associations identified for each mRNA (as a kind of summary of the text).
Table 1 - try to organize the data in the "Cohorts" column for greater coherence. Also in the Link column - replace with Authors and pass only the Name Surname et al. (no year).
Line 253 – Four instead of 4
Fig 2-5 – do the colors of the points have any significance?
At the end of the discussions, add a paragraph in which you mention the importance of this study and its limitations.
Add the authors' contributions.
Formulate the references according to the instructions.
Is reference 16 absolutely necessary (its importance does not emerge, except for self-citation purposes, the same for reference 17)?
Author Response
Thank you very much for your interest in our manuscript and comments, we really appreciate it. We have tried to answer your remarks and made clarifications in manuscript. All changes are highlighted in yellow.
All references removed at the end of the sentence.
We have removed the publication years of articles cited in the text.
EEG symptoms were changed to EEG signs.
In subchapter 3.2, we added a table that would summarize the data presented earlier.
We have tried to organize the data in the cohort column in Table 1, and the Link column has been modified and renamed.
The drawings were made using the Tissue Atlas database. To collect information in this database, autopsy materials from six sources were used, the color of the dots indicates the number of the source from which the biological material was taken.
The "discussion" section has been modified, additional information has been added.
Authors' contributions. have been added.
References have been modified.
We tried to take into account all your comments.
Sincerely,
Authors